# Notes on the Summer Life History Traits of the Non-Native Pumpkinseed (*Lepomis gibbosus*) (Linnaeus, 1758) in a High-Altitude Artificial Lake

**Alexandra S. Douligeri** [1], **Athina Ziou** [1], **Athanasios Korakis** [2], **Nikolaos Kiriazis** [2], **Nikolaos Petsis** [2], **George Katselis** [1] **and Dimitrios K. Moutopoulos** [1,*]

[1] Department of Fisheries & Aquaculture, University of Patras, 30200 Mesolongi, Greece; alexandra.douligeri@gmail.com (A.S.D.); athinaziou@gmail.com (A.Z.); gkatselis@upatras.gr (G.K.)

[2] Management Unit of Northern Pindos National Park, N.E.C.C.A., 44007 Ioannina, Greece; akorakis@hotmail.com (A.K.)

[*] Correspondence: dmoutopo@upatras.gr

**Abstract:** In the present study, the biology of the pumpkinseed *Lepomis gibbosus* in the artificial lake of Aoos, located in northwestern Greece, was investigated. The samplings of the pumpkinseed were conducted from the shore using a portable electrofishing device over a 4-month period (July 2021–October 2021). A total of 581 specimens were caught, with an average length of 62 mm. The sex ratio of female to male was estimated to be 1.0:1.7, and the percentage of mature specimens was estimated for all of the months to be above 52%, matching the highest percentage in July (57.4%). The b value of the length–weight relationship ranged from 3.16 in September to 3.31 in July. The value of the L∞ and K was estimated to be equal to 119 mm and 0.36 years$^{-1}$, respectively, and the value of φ′ was equal to 3.707. The total mortality was estimated to be equal to 1.63 ± 0.48 y$^{-1}$ ($R^2$ = 0.96), and the natural and fishing mortalities were 0.83 and 0.80, respectively. The maximum age was 6 years, and the theoretical maximum age was 8 years. In the current study, the value of the L∞ was estimated to be near the European average but significantly lower than the North American one, whereas the value of the K was slightly higher than the European average. The small size of the specimens obtained in Aoos Springs was most likely owed to the combined impact of the investigated lake's high altitude and low food availability, resulting in a limited factor for species expansion.

**Keywords:** growth parameters; condition factor; spatial distribution; Greece





## 1. Introduction

Non-native species have migrated, survived, and reproduced across a range of habitats and are putting strain on native fauna. Such species are known to cause adverse environmental, economic, and social impacts, such as alterations in the populations of native species, the transmission of diseases, and significant irreversible changes in the natural environment [1]. Consequently, international agreements, initiatives, regulations, and conservation strategies have been developed to prevent their spread and timely eradicate and manage established populations, thereby preserving biodiversity [2]. Inland fauna in Greece displays one of the highest degrees of endemicity in Europe [3,4]. However, these populations are under severe pressure due to reduced rainfall and an increase in water temperature pollution and human activities, which are further enhanced by the introduction and spread of non-native species [5–7].

The high-altitude artificial lake of Aoos has evolved into an important mountainous aquatic ecosystem with an abundance of native aquatic organisms. In contrast, certain non-native fish species have been introduced into the system, including the pumpkinseed *Lepomis gibbosus* (Linnaeus, 1758) and Prussian carp *Carassius gibelio* (Bloch, 1782), which are considered among the most threatening species for global biodiversity [8] but also

for the lake's aquatic fauna. Pumpkinseed is considered a non-native species capable of causing disturbances within the habitat and fish fauna of inland waters that have been introduced [9].

The first introduction of the pumpkinseed to Europe, and specifically to France, is believed to have occurred in 1877 [10] through river channels. The purpose and history of the species introduction into the inland waters of European countries are somewhat unclear. As the pumpkinseed can be seen as an ornamental species, it was introduced and stocked in small garden ponds and aquariums, from which it was either deliberately released or accidentally escaped into inland waters, which played a key role in the spread of the species [11]. Another reason is its use as bait by recreational fishers [12]. The pumpkinseed was first reported in Greece in 1885 [13], and the first settled population was recorded approximately 100 years later in the Aliakmonas River in Macedonia (Northwest Greece) [14]. It was recently introduced into the artificial lake of Aoos during the middle of the preceding decade [15].

Pumpkinseed prefers shallow waters, with little water movement and enough vegetation, and the temperature it lives in ranges from 4 °C to 30 °C [16]. It thrives and reproduces in a variety of habitats [17], and its presence may have a major impact on the richness of the ecosystems in which it settles [18]. The breeding season of pumpkinseed begins when the water temperature rises, usually around 20 °C, although the duration and specific timing may vary depending on the region [19]. Males are capable of breeding with several females in succession. After spawning, the male guards the eggs in a nest and subsequently protects juveniles until they absorb their yolk sac [20]. The species typically reaches sexual maturity at around 1–2 years of age [20]. With respect to feeding habits, the pumpkinseed primarily feeds on worms, crustaceans, and insects. Additionally, it consumes small fish, fish eggs, and other vertebrates [21]. Distinct differences exist between North American populations, where the pumpkinseed is native, and European populations [22]. In Europe, the average length and growth rate of adult pumpkinseeds tend to be smaller compared to those in America. Moreover, the average asymptotic length (L∞) is higher in American populations than in European ones. Up to this point, the pumpkinseed has been documented in 32 lentic ecosystems across Greece [23]. However, information regarding its biology is limited to a survey conducted in the artificial lake of Kerkini two decades ago [12] and, more recently, a study investigating its abundance in Lake Volvi over a three-year period [9].

The purpose of this study is to investigate the biology of the pumpkinseed in the artificial lake of Aoos by assessing its biometric characteristics, robustness, and life history parameters, including age estimation, asymptotic length, and mortality. Sampling was employed during the breeding season and within the breeding habitats of the species because the sampling design was set to manage the population of the species by selectively removing sexually mature specimens found in the nests [24]. In the context of an ecosystem-based approach, the findings of the present study were compared with those of the North American, where the species is native, and European populations, where the species is non-native, in order to investigate the area's effect (the latitude and altitude) on the life history traits of the studied species. This conjunctive analysis holds particular significance due to the fact that the studied lake is a high-altitude alpine-type lake (1343 m) characterized by strong temperature fluctuations [25] that impact the survival of the organisms within it. The findings from this research will contribute to the effective management of the pumpkinseed population in the studied system.

## 2. Materials and Methods

### 2.1. Study Area Sampling

The artificial lake of Aoos is an "alpine-type" ecosystem located at an altitude of 1343 m within the mountainous area of Northern Pindos (Figure 1). It covers an area of 11.5 km$^2$ and has a capacity of $260 \times 10^6$ m$^3$, with a maximum depth reaching 80 m. The system is oligotrophic, characterized by an annual temperature fluctuation ranging from 4 °C to 26 °C, while the surface level undergoes an approximate 10 m variation per year [25].

The system is monomictic when there is no ice formation, changing to dimictic when ice is formed [26]. The water column is thermally structured during summer, whereas it appears to be quite uniform from the surface to the bottom throughout the rest of the year [25]. Water temperature annually ranges from 4 to 6 °C in March to 5 to 21 °C in July, with the formation of thermal layers occurring at depths of 5–15 m during the summer season [25]. Mountain springs and river runoff discharged from adjacent mountains are limited, and the system assembles the characteristics of a natural lake [26]. The fish fauna of the artificial lake of Aoos consists of both native species, such as *Salmo farioides*, Karaman 1938, *Barbus prespensis*, Karaman 1924, *Squalius* sp. Aoos ('unnamed and undescribed taxa', according to [27]), and *Alburnoides bipunctatus*, Bloch 1782, as well as introduced species, such as *Oncorhynchus mykiss*, Walbaum 1792, *Cyprinus carpio*, Linnaeus 1758, *Carassius gibelio*, Bloch 1782, and *Acipenser gueldenstaedtii* Brandt and Ratzeburg 1833, which are confirmed by recreational fishers [28].

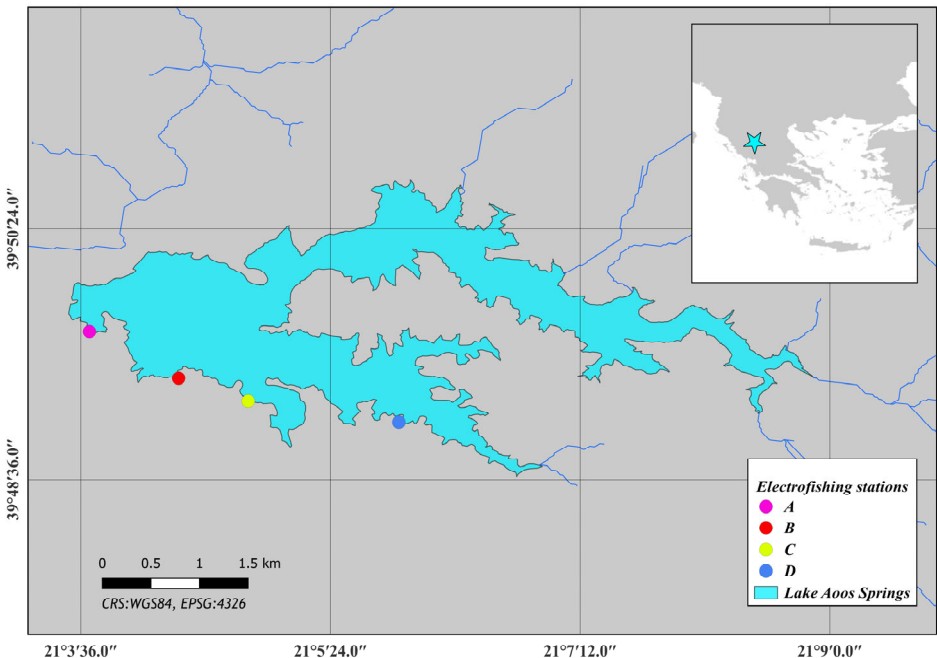

**Figure 1.** Sampling sites in the artificial lake of Aoos.

Experimental electrofishing from the coast was applied around the lake during the morning hours with a frequency of approximately every ten days over a four-month period (July 2021–October 2021). The device used was a Hans-Grassl GmbH battery-powered backpack, Model IG200-2, DC (pulsed), 1.5 KW output power, 35–100 Hz, max. 850 V (Schönau, Germany). Electrofishing was applied at four stations (A, B, C, and D) in the southern part of the lake in the creeks with shallow waters and based on the presence of nests at these points (Figure 1). In particular, stations C and D are places where nesting of the species was recorded, while in stations A and B, which have similar characteristics of the substrate (modified coastline with rip-rap), nesting was not recorded.

### 2.2. Data Collection

The specimens were preserved in formalin and transported to a laboratory where each specimen was measured for the total and standard length with a precision of 0.1 cm. The gross weight was also recorded, and the liver and gonads were removed and weighed. In the case of mature specimens, their sex was also identified. All weight measurements were taken using a balance with an accuracy of 0.01 g. Total length data were used to separate length classes every 5 mm for the entire sample period, and then we estimated the length–weight relationship in the total sample monthly and by sex by applying the following equation: $W = a\,TL^b$, where W is the gross weight, TL is the total length, and coefficients

a and b are the intercept of the curve on the weight axis and the slope of the equation, respectively. Coefficient b usually takes values from 2 to 4, both inter- and intra-specifically, when there are records from different regions, seasons, and years [29]. Prior to the use of a statistical test, the distributions of the length data per sex and season were examined for whether or not they deviated significantly from the normal distribution by using the Shapiro–Wilk test [30]. The comparison of the mean total length between the months was performed with co-variance analysis (ANOVA, $p < 0.05$), whereas the comparison of the sex with the seasons (Summer–Autumn) was performed with a Student's *t*-test ($p < 0.05$). The comparison of the respective distributions between the two different seasons was performed with Kolmogorov–Smirnov analysis (K-S test) [30]. The comparison of the length–weight relationship with months was performed using an analysis of co-variance (ANCOVA, MS Office Excel 2019).

To determine the breeding season of the species, the gonads of sexually mature specimens were removed and weighed. The gonadosomatic index was then estimated using the equation [31]: GSI = (GW/W) × 100, where GW is the weight of the gonad measured in gr, and W is the total weight of the fish. The index was estimated separately for each sex and month to identify the main breeding season of the species. Additionally, the sex ratio was estimated to provide information on the prevalence of each sex, the season, and sampling location.

The length data were utilized to calculate the growth rate (K) and asymptotic length (L∞) using the ELEFAN I method [32] from FISAT II software [33]. The smallest length class was set to 20 mm, and the class intervals were set to 5 mm. To determine the best fit, the fit index, Rn [32], was applied to the length–frequency data set. Age, $t_0$, was estimated using Pauly's equation [34]: $Log(-t_0) = -0.3922 - 0.2752 * LogL∞ - 1.038 * LogK$, where L∞ and K represent the growth parameters estimated before. The growth efficiency index was calculated using the equation [35]: $φ' = LogK + 2 * LogL∞$.

Parameters L∞ and K were used to estimate the total mortality (Z) by employing the length-converted catch curve in the FISAT II software [33]. Natural mortality (M) was estimated using parameters L∞ and K, according to the empirical equation of [32]: $Log(M) = -0.0066 - 0.2790 * Log(L∞) + 0.6543 * Log(K) + 0.4634 * Log(T)$, where T is the average annual temperature of the lake, which, in the present study, was T = 13 °C [26]. Fishing mortality (F) was estimated as the difference between total mortality (Z) and natural mortality (M), $F = Z - M$.

To determine the age of the fish, scales were carefully removed from the bottom of the pectoral fins and photographed using an electron microscope with a resolution of 500×. Subsequently, a photo editor was employed to measure the radius and distances between rings to confirm that they were annual. The length classes were then grouped by age, and the total length-at-age keys were constructed using 5 mm length intervals. Specimens at age 0 were identified as those that had not yet completed the formation of the first ring. To estimate the maximum age ($t_{max}$) of the species in the lake, von Bertalanffy's equation [36] was used: $L_{max} = L∞ * (1 - e(-K(t_{max} - t_0)))$, where $L_{max} = 0.95 * L∞$.

A literature review has also been conducted on growth studies of pumpkinseed found in North America, where the species is native, and Europe, where the species is non-native, and specifically, the relationships among the life history traits, as well as between these traits and the age–class identified in the above-mentioned areas, and with the area's effect (latitude, altitude).

## 3. Results

### 3.1. Estimation of Life History Traits

The length and weight were measured for a total of 581 specimens of pumpkinseed, where 364 specimens were collected during the summer (July–August 2021), and 217 were collected during the autumn (September–October 2021) samplings. The highest number of specimens (310) was collected in August, while the lowest number (54) was collected in July (Table 1). The length ranged from 24 mm to 101 mm, with the mean length of all the

specimens estimated to be 62 mm (SD: 13 mm). In the summer, the mean length was 64 mm (SD: 13 mm), while in the autumn, it was 58 mm (SD: 15 mm). The Shapiro–Wilk test for the length data by season and sex showed that their distribution did not deviate significantly ($p < 0.05$) from the normal distribution. The comparison of the length frequency distributions between seasons exhibited no significant (K–S = 0.71, $p > 0.05$) differences. A total of 108 females and 193 males were identified, while 280 specimens did not have detectable gonads, either due to the release of reproductive material or because they had not yet reached the age of first maturity. The mean length of the males (mean: 70 mm, SD: 10 mm) was significantly (Student's *t*-test, $p < 0.05$) longer than that of the females (mean: 67 mm, SD: 8 mm). The specimens whose sex was not identified had a significantly (Student's *t*-test, $p < 0.05$) shorter mean length (mean value: 54 mm, SD: 11 mm) compared to both sexes. In terms of size classes, most of the males were found in the 60–80 mm range, while the females were predominantly in the 50–70 mm range (Figure 2).

**Table 1.** Descriptive statistics and parameters of total length and length–weight relationship, $W = aTL^b$ [weight (in g) and length (in cm)] of pumpkinseed *Lepomis gibbosus* by month, season, and for all months for the samples caught in the artificial lake of Aoos in 2021. $R^2$ is the coefficient of determination.

| Month/Season | | N | TL | | | | $W = aTL^b$ | | |
|---|---|---|---|---|---|---|---|---|---|
| | | | Min | Max | Mean | SD | a-Value | b-Value | $R^2$ |
| Month | July | 54 | 42 | 90 | 63 | 11.9 | 0.009 | 3.306 | 0.990 |
| | August | 310 | 44 | 101 | 64 | 13 | 0.011 | 3.212 | 0.983 |
| | September | 160 | 30 | 95 | 63 | 15 | 0.013 | 3.156 | 0.982 |
| | October | 57 | 24 | 71 | 44 | 14.8 | 0.012 | 3.174 | 0.994 |
| Season | Summer | 364 | 42 | 101 | 64 | 13 | 0.011 | 3.233 | 0.984 |
| | Autumn | 217 | 24 | 95 | 58 | 15 | 0.012 | 3.194 | 0.986 |
| | TOTAL | 581 | 24 | 101 | 62 | 13 | 0.012 | 3.192 | 0.985 |

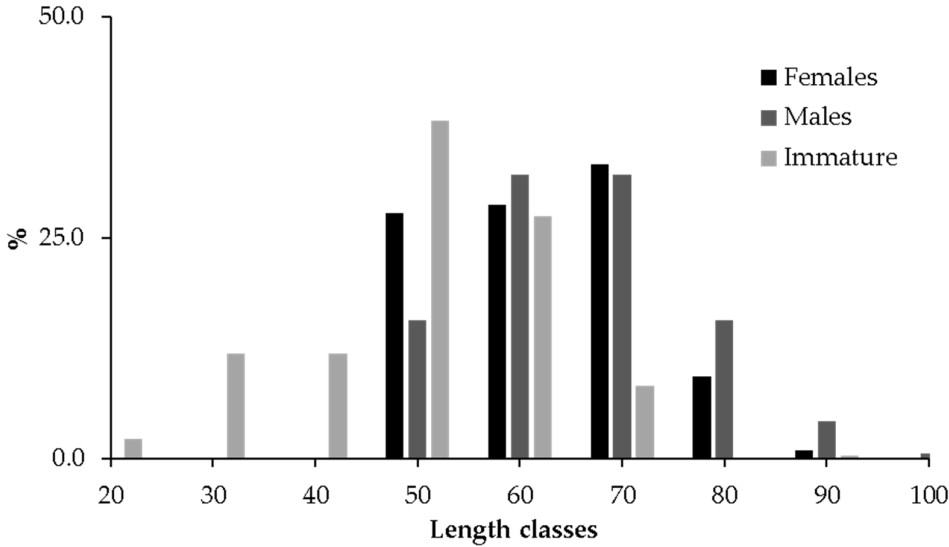

**Figure 2.** Length class per sex of pumpkinseed *Lepomis gibbosus* specimens caught from July 2021 to October 2021.

The estimation of the length–weight relationship, conducted for the entire sampling period on a seasonal and monthly basis, revealed that in all of the cases, the weight exhibited a significant (all $R^2$ values were greater than 0.982) positive allometric relationship with the length, with slope b (3.15) being significantly (Student's *t*-test, $p < 0.05$) higher than the isometric growth (Table 1). The monthly variability of slope b ranged from 3.156 in September to 3.306 in July, while parameter a ranged from 0.009 in July to 0.013 in September

(Table 1). When comparing the length–weight relationship between the sexes, there was no significant (ANCOVA, $p > 0.05$) difference observed for a and b. However, when comparing parameters a and b by month, significant (ANCOVA, $p < 0.05$) differences were found for each pair of months, except for parameter a in July, which did not significantly (ANCOVA, $p > 0.05$) differ from the other three months.

The sex ratio was 1.0:1.7, female to male. The mature and immature specimens were almost equal throughout the entire sampling period, with the exception of October, when the majority (78.9%) were immature. The highest proportion of reproductive maturity was observed in July, with a percentage of 57.4%.

The females exhibited significantly (ANOVA test; $p < 0.05$) higher values of the gonadosomatic index compared to the males (Figure 3A). Among the sexes, the highest gonadosomatic index value was estimated in July, reaching 4.67 for the females and 1.41 for the males. Conversely, the lowest values were recorded in September (0.49 and 0.34, respectively). In October, there was a slight increase in the gonadosomatic index for both of the sexes, with an increment of 0.05 for the males and 0.11 for the females. Analyzing the monthly fluctuations of the condition factors in relation to the gonadosomatic index, the condition factors reached their maximum values when the gonadosomatic index exhibited its minimum value (Figure 3B). Specifically, the gonadosomatic index exhibited its highest value in July, while condition factors were at their lowest. In contrast, during September, all of the condition factors marked an increase, while the gonadosomatic index reached its minimum value.

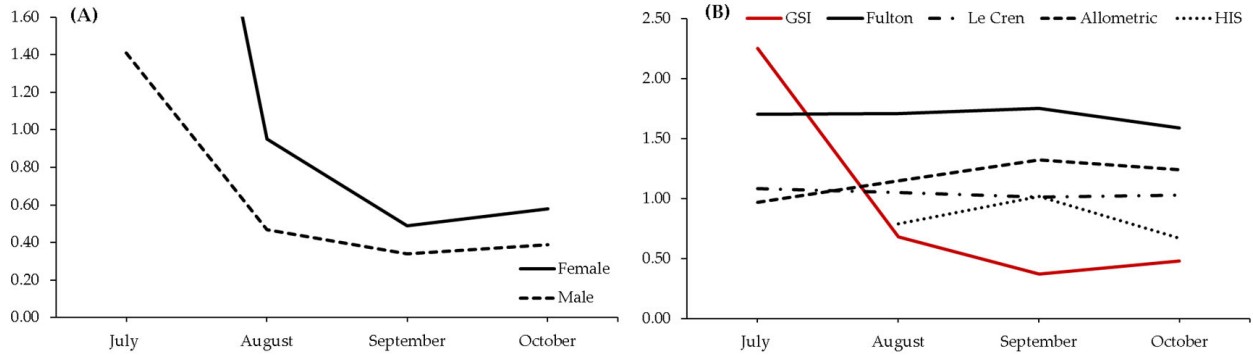

**Figure 3.** (**A**) Monthly variation of the gonadosomatic index (GSI) by sex. (**B**) Monthly variation of the four condition factors studied (Fulton, Le Cren, Allometric, HIS) in relation to the variation of the gonadosomatic index (GSI) for pumpkinseed in the artificial lake of Aoos.

The values of the L∞ and K were an estimated 119 mm and 0.36 years$^{-1}$, respectively. The index, Rn, was 0.691. The value of $t_0$ was an estimated $-0.31$ years. The growth efficiency index, φ′, was estimated to be equal to 3.707. Total mortality, Z, was estimated to be equal to $1.63 \pm 0.48$ y$^{-1}$ ($r^2 = 0.956$), with a 95% confidence interval between 1.15 and 2.12 y$^{-1}$. The natural mortality, M, was estimated to be equal to 0.829 for a mean annual water temperature of 13 °C. The fishing mortality, F, and the exploitation rate, E, were estimated to be 0.80 and 0.49, respectively.

Length classes were constructed in 5 mm increments, ranging from a minimum length of 20–25 mm to a maximum length of 100–105 mm, covering an age range of 0 to 6 years (Table 2). The largest percentage of specimens was observed between 55 mm (19.8%) and 60 mm (17.4%), which were between 0 and 2 years old. At age 0+, there is a wide range of lengths, starting from 20 mm and extending up to 65 mm. At 50 mm, there is an overlap with the specimens aged 1-year-old. The third largest percentage of specimens (14.6%) was found at 70 mm, representing the age range of 1 to 3 years. The majority of the specimens collected were 1 year old (35.3%), followed by 0 (25.3%) and 2 (23.9%) years. Older specimens accounted for less than 9% of the total. Additionally, only one specimen of 6 years was found, measuring over 100 mm in length.

**Table 2.** Length by age relationship of pumpkinseed in the artificial lake of Aoos.

| Length Classes (mm) | 0 | 1 | 2 | 3 | 4 | 5 | 6 | Total | % |
|---|---|---|---|---|---|---|---|---|---|
| | | | | Age | | | | | |
| 20–25 | 1 | | | | | | | 1 | 0.2 |
| 25–30 | 5 | | | | | | | 5 | 0.9 |
| 30–35 | 17 | | | | | | | 17 | 2.9 |
| 35–40 | 16 | | | | | | | 16 | 2.8 |
| 40–45 | 16 | | | | | | | 16 | 2.8 |
| 45–50 | 17 | | | | | | | 17 | 2.9 |
| 50–55 | 37 | 15 | | | | | | 52 | 9.0 |
| 55–60 | 26 | 87 | 2 | | | | | 115 | 19.8 |
| 60–65 | 9 | 81 | 11 | | | | | 101 | 17.4 |
| 65–70 | 3 | 18 | 45 | 3 | | | | 69 | 11.9 |
| 70–75 | | 4 | 60 | 21 | | | | 85 | 14.6 |
| 75–80 | | | 19 | 13 | 4 | | | 36 | 6.2 |
| 80–85 | | | 2 | 10 | 12 | 1 | | 25 | 4.3 |
| 85–90 | | | | 3 | 9 | 3 | | 15 | 2.6 |
| 90–95 | | | | | 3 | 3 | | 6 | 1.0 |
| 95–100 | | | | | | 4 | | 4 | 0.7 |
| 100–105 | | | | | | | 1 | 1 | 0.2 |
| Total | 147 | 205 | 139 | 50 | 28 | 11 | 1 | 581 | |
| % | 25.3 | 35.3 | 23.9 | 8.6 | 4.8 | 1.9 | 0.2 | | |

The mean length by the age curve for the pumpkinseed reveals a noticeable increase in the mean length per age, ranging from 47 mm in the 0-year-olds to 90 mm in the 5-year-olds. Additionally, there was a specimen measuring 101 mm in length, identified as age 6 (Table 2). The maximum age ($t_{max}$) of the pumpkinseed in the artificial lake of Aoos was estimated equal to 8 years.

### 3.2. Metanalytic Approach of Life History Traits

According to the results of the current study, the estimated value of parameter b in the length–weight relationship (3.190) was the second highest compared to the corresponding estimates from other European freshwater systems, where the species has been introduced, indicating a relatively good condition in our study area. Specifically, the b parameter varied from 2.980 in the Segura River, Spain [37], to 3.491 in a river basin in Turkey [38] (Appendix A, Table A1).

Regarding the comparisons of life history parameter estimates, the value of the L∞ in the present study lies within the European mean (120.7 mm) (Appendix A, Table A2), but it is lower than the average value estimated in North American freshwater systems (188.4 mm), where the species is native. Specifically, in European inland waters [22], the values of the L∞ range from 81.3 mm in the Tapada Pequena reservoir in Portugal to 168.3 mm in a section of the Danube River in Slovakia. In North America, the values of the L∞ range from 125.3 mm in Lower Beverly Lake, Canada, to 292.5 mm estimated in various systems studied in the Minnesota region, USA [22]. In Greece, only one survey of the species biology has been conducted in the artificial lake Kerkini, with the L∞ value (136.5 mm) being the second highest in Europe. The value of parameter K was estimated in our area to be 0.36 $yr^{-1}$, slightly higher than the mean value estimated in European freshwater systems (0.33 $yr^{-1}$). In the artificial lake Kerkini, the value of K is 0.13 $yr^{-1}$, while in North America, the K ranges from 0.13 $yr^{-1}$ in Upper Beverly Lake, Canada, to 0.63 $yr^{-1}$ in surveys conducted in various water bodies in Delaware, USA.

In the present study, a total of seven age classes were identified, which is estimated to be the predominant age distribution for pumpkinseed in European freshwater systems (Figure 4). However, there were a few exceptions. The Tapada Grande reservoir in Portugal, as well as the Dabas and Cottesmore waterholes in Hungary and England, respectively,

exhibit five age classes. At the Tapada Pequena reservoir in Portugal, a section of the Danube River in Slovakia, and the artificial lake Kerkini in Greece, six age classes were determined. The Divor reservoir in Portugal had the highest number of age classes, reaching up to eight age classes. In North America, where the species is native, in certain systems, the number of estimated age classes exceeded seven and reached up to ten [22].

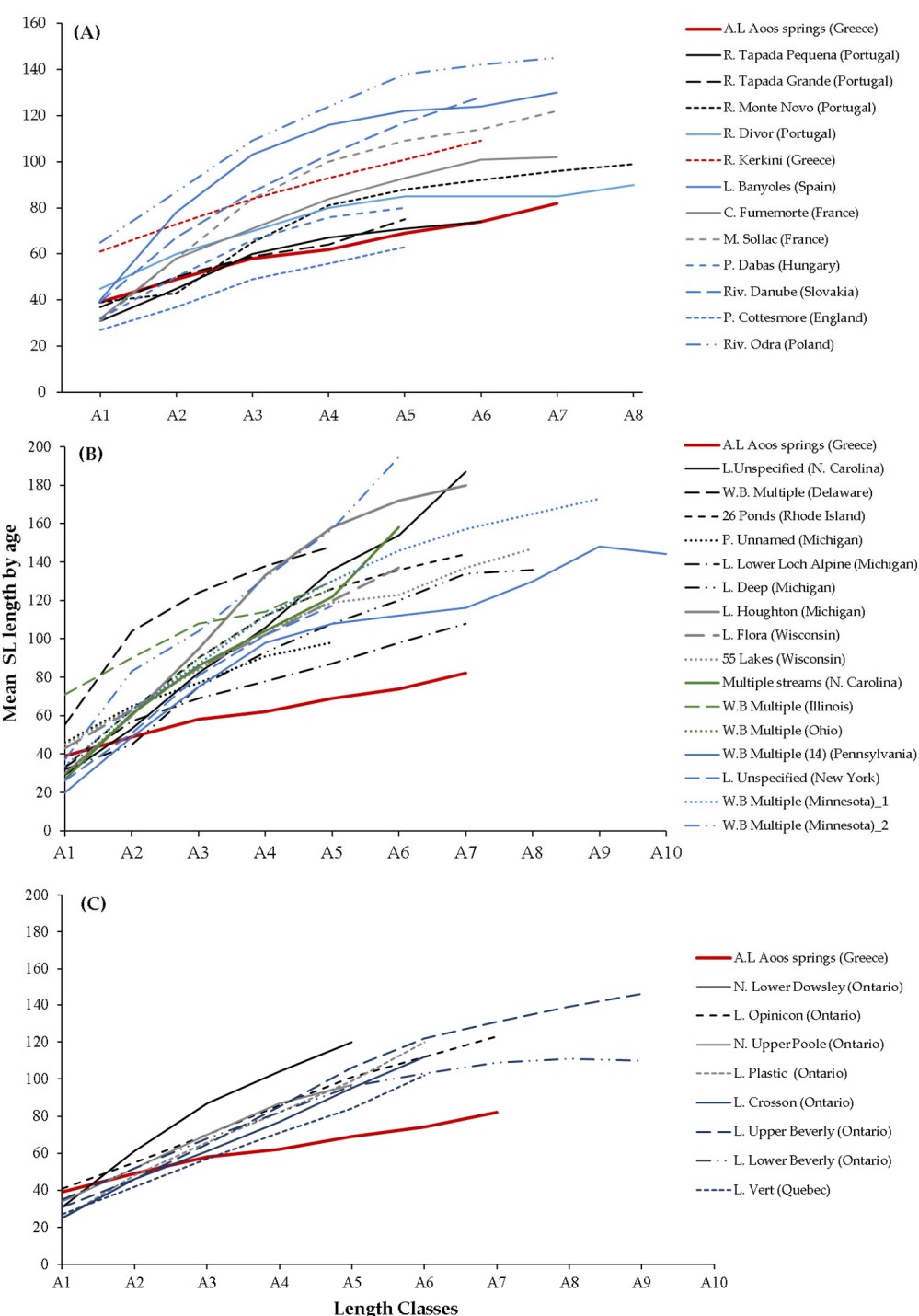

**Figure 4.** Relationship of the mean standard length by age class of pumpkinseed *Lepomis gibbosus* in different freshwater systems: (**A**) in Europe, (**B**) in North America, and (**C**) in Canada. R: Reservoir, A.L: Artificial Lake, L: Lake, C: Canal, M: Marsh, P: Pond, Riv: River, W.B: Water Bodies.

To compare the length–age relationships across the various studied ecosystems, the standard length of the specimens was utilized. The mean length for each age class in the

present study was estimated to be among the smallest for European freshwater systems (Figure 4A) and smaller than the mean length estimated in populations studied in the United States (Figure 4B) and Canada (Figure 4C). The specimens in the first age class in our study area exhibited an average standard length of 39 mm, whereas the corresponding average length for the same age group in North America (USA and Canada) was estimated to be 35 mm. The differences in the average standard lengths were more pronounced in the subsequent age groups, with the mean lengths in North America being greater.

The Le Cren condition factor estimated for *L. gibbosus* in European freshwater systems ranges from 0.88 in Lake Cottesmore School, England [39] to 1.19 in Lake Grand-Lieu, France [39] (Table 3). The robustness value estimated in the present study exhibited the second highest value compared to those estimated from other European freshwater systems (Table 3). In contrast, the gonadosomatic index value was lower than the mean value estimated in other European freshwaters [39,40]. Specifically, the gonadosomatic index values in other systems range from 3.14 in samples collected in June from the Slangebeek stream, Belgium, to 11.36 in samples collected in May from Lake Grand-Lieu, France (Table 3).

**Table 3.** Gonadosomal index (GSI) and Le Cren condition factor ($K_R$) values of pumpkinseed *Lepomis gibbosus* in European freshwater systems.

| Region | Country | Month | GSI | $K_R$ | Year | Reference |
|---|---|---|---|---|---|---|
| Artificial lake of Aoos | Greece | July | 4.67 | 1.08 | 2021 | Present study |
| 19 water bodies | England | June | 7.70 | 1.02 | 2003–04 | [40] |
| Lake Einedammen | Norway | July | 7.22 | 0.91 | 2006 | [39] |
| Lake Schoapedobbe | Netherlands | June | 10.04 | 1.00 | 2006 | [39] |
| Lake Meeuwven | Netherlands | June | 8.78 | 0.91 | 2006 | [39] |
| Lake Cottesmore School | England | June | 9.80 | 0.88 | 2006 | [39] |
| Stream Batts Bridge | England | June | 4.13 | 0.97 | 2006 | [39] |
| Isle of Wight | England | June | 6.71 | 1.07 | 2005 | [39] |
| Lake Webbekomsbroek | Belgium | June | 6.10 | 1.04 | 2006 | [39] |
| Stream Slangebeek | Belgium | June | 3.14 | 1.05 | 2006 | [39] |
| Marsh Brière | France | May | 11.20 | 1.04 | 2006 | [39] |
| Lake Grand-Lieu | France | May | 11.36 | 1.19 | 2006 | [39] |

The theoretical maximum value of the $t_{max}$ for pumpkinseed in our study area was estimated to equal 8 years, which is lower than the mean value estimated in other European freshwater systems ($t_{max}$ = 10 years: Table A2), as well as in North American lakes (tmax = 16 years: Table A2). In the majority of North American freshwater systems, the $t_{max}$ was estimated to be higher than the mean value estimated in European populations. This highlights the fact that although the species is non-native to these ecosystems, it seems to have adapted to a satisfactory extent in European ones. Additionally, the relatively earlier maximum age observed in pumpkinseed populations in the artificial lake of Aoos compared to other systems is likely attributed to the wide temperature ranges and low nutritional environment that the species encounters in an alpine, high-altitude lake ecosystem.

The relationship between the L∞ and K of pumpkinseed from the inland waters of Europe and North America based on the literature [22] depicted a significant ($p < 0.05$) negative linear regression with the smallest values of the L∞ and the highest ones of the K estimated from the present study (Figure 5A). The relationship between the L∞ and the latitude of each system found with the corresponding estimates of the life history traits of pumpkinseed depicted a significant ($p < 0.05$) positive linear regression with high L∞ values estimated at higher latitudes (Figure 5B).

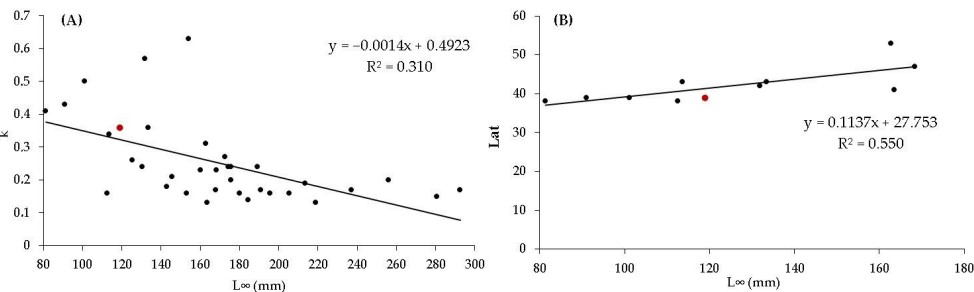

**Figure 5.** (**A**) Relationship between asymptotic length (L∞) and growth rate (K) of pumpkinseed *Lepomis gibbosus* from inland waters of Europe and North America based on the literature [24]. (**B**) Relationship between estimated asymptotic length (L∞) of *Lepomis gibbosus* with latitude in European inland water systems [24]. Red circle indicates the estimates from the present study.

## 4. Discussion

The present study aimed to investigate the biology of pumpkinseed within a high-altitude artificial lake. The analysis of the data could be valuable for the management of the species population in the artificial lake of Aoos, considering its non-native nature and impact on the lake's habitat and fish fauna [15]. The findings of this study also provide important insights and serve as a valuable update to the initial study conducted in Greek inland waters 20 years ago [12].

The observed numerical predominance of males over females (1.7:1.0) was anticipated due to the specific sampling method employed during the breeding season and within the breeding fields of the species [24]. During this period, males construct nests and assume the responsibility of guarding and protecting the fertilized eggs, which often leads to the exclusion of females from the nesting area [41]. The analysis of condition factors, as well as the estimation of the gonadosomatic index, revealed a distinct pattern for our study area. It was observed that the pumpkinseed displayed its highest robustness in September, whereas it was lowest during summer. Despite the lack of samples during spring, when the species also reproduce, the highest values of the gonadosomatic index observed in both male and females during July is supported by other studies [12,42,43]. Furthermore, it aligns with the period when the system reaches the highest water temperature (21 °C) [26], which promotes the initiation of species reproduction [44]. The deposition of reproductive material during the summer is further supported by the fact that over 52.0% of the specimens were sexually mature in all of the months except for October when the majority were immature (78.9%).

The above observations are further supported by comparing the robustness indices with the gonadosomatic index (Figure 3B). The monthly variation of the robustness indices exhibited an inverse pattern compared to the gonadosomatic index. Specifically, the gonadosomatic index reached its maximum value in July, while the condition factors exhibited their lowest values during this month. This can be attributed to the fact that during the reproductive period in summer, freshwater fish allocate a significant amount of energy towards the development of their gonads, which affects their overall body condition and robustness [45,46]. This is further reinforced by the observation that in September when all of the condition factors showed an increase, the value of the gonadosomatic index was at its minimum. In contrast, a different pattern was observed in the artificial lake, Kerkini [12], where the gonadosomatic index exhibited a consistent pattern of increasing and decreasing values throughout the months.

The estimated total mortality was 1.63, with natural mortality estimated to be 0.829. These mortality estimates are the first for the *L. gibbosus* in Mediterranean inland waters. The standard deviation of natural mortality, which is equal to 0.245 [47], aligns with the standard deviation of the Z value. This is to be expected given that pumpkinseed is not a target of recreational fishing in the study area, as well as commercial fishing in the country's inland waters, where professional fishing is permitted [48], and the species is primarily used as a decorative fish in aquariums [11].

The growth rate of fish is influenced by external factors such as water temperature and food availability, as well as internal factors related to the genetic characteristics of the organisms [45]. The unique characteristics of the artificial lake of Aoos, being an alpine-type lake located at a high altitude (1343 m) with significant temperature fluctuations (ranging from 4 °C to 26 °C) and the potential for ice formation under extreme conditions [26], create an ecosystem that hampers the growth of non-native species like the pumpkinseed. Water temperature plays a crucial role in food intake and growth, as lower temperatures lead to decreased growth rates [45]. Moreover, the system is characterized as an oligotrophic lake [26], lacking in essential organisms that the pumpkinseed feeds on, such as benthic invertebrates [15]. Consequently, these conditions favor the dominance of the small-sized individuals of the species, as they require less food and energy to attain their needs [49]. This might suggest that the species follows a k-type survival strategy [50], characterized by small body size, high growth rate, and shorter lifespan. This hypothesis is further supported by the positioning of the present estimates in the left region of the $L\infty$ and K relationship diagram in comparison to estimations from other inland water systems in Europe and North America (Figure 5A). Thus, pumpkinseed might be adapted to the extreme conditions of our study area by adopting a k-type strategy. The nests that *L. gibbosus* creates in the lakes range from 40 to 125 cm in diameter [51], while in the studied system, their diameter ranges from 25 to 45 cm, which is related to the size of the male individuals of the species that build the nests [52].

With respect to the low values of the gonadosomatic index, the lack of samples during spring might bias the corresponding values, and this is the biggest drawback of the study. Alternatively, we might assume that the low value of the gonadosomatic index estimated in our study area is likely associated with the observed low growth rate, as discussed earlier, which is influenced by the low temperatures characteristic of high-altitude alpine lakes like the studied system. The low values of the gonadosomatic index might also indicate that the pumpkinseed population in the high-altitude artificial lake of Aoos was relatively low, possibly due to the limited biomass of its prey, such as benthic invertebrates. Although pumpkinseed exhibited quite flexible feeding habits in response to changes in the availability of food resources or environmental conditions along the invasion process [53,54], it seemed that in the studied system, the prey of this species was at a low availability. In general, the balance of pumpkinseed biomass in lake ecosystems is likely regulated by trophic interspecific competition for available food resources [55–57]. Predatory fish species like pike (*Essox lucius*) and pike-perch (*Sander lucioperca* (Linnaeus, 1758)) are known to prey on pumpkinseed [58], and cannibalism has also been observed in certain cases [59].

Latitude consists of a limiting factor in fish growth, with organisms living at higher latitudes tend to be larger and more robust than those near the equator in order to better maintain their body temperature [60]. This pattern appears to apply to pumpkinseed in European freshwater systems [61,62] (Figure 5B) and goes against the conclusions of this study because the temperatures recorded in the studied system are low and similar to those in areas at higher latitudes but with more accessible food resources (Table A2) [63]. Therefore, the smaller size of the pumpkinseed in the studied system is likely a result of the nutritional constraints they experience, which hinder their growth and size increase.

## 5. Conclusions

According to the results of the present study, the pumpkinseed, despite being a non-native species in the artificial lake of Aoos, has not managed to reach the sizes observed in other European systems. This can be attributed to the unique biotic and abiotic characteristics of this high-altitude artificial lake. There is a link between growth and temperature, with organisms below their temperature optimum predicted to develop faster with increasing temperature, while those above their temperature optimum are expected to grow slower [64]. Given that the pumpkinseed exhibited a temperature range of 4 °C to 30 °C for optimal growth [16] and the studied system experiences temperatures below 12 °C during

a great part of the year [26], it is possible that an increase in temperature, possibly due to climate change, could favor the growth of the species in the future.

**Author Contributions:** Conceptualization, G.K. and D.K.M.; methodology, D.K.M.; formal analysis, A.S.D.; investigation, A.S.D. and A.Z.; resources, A.S.D., A.Z., A.K., N.K. and N.P.; data curation, A.S.D. and D.K.M.; writing—review and editing, A.S.D., A.K., G.K. and D.K.M.; supervision, D.K.M.; project administration, N.K., A.K., N.P. and G.K.; funding acquisition, G.K. All authors have read and agreed to the published version of the manuscript.

**Funding:** This study was performed in the framework of the research project "implementation and evaluation of management measures for the fish fauna of the artificial lake of Aoos with emphasis on the sustainable management of the non-native species", funded by the Management Agency of the Northern Pindos National Park-Now Management Unit of the Northern Pindos National Park under the Natural Environment and Climate Change Agency (N.E.C.C.A.), through the Operational Program Transport Infrastructure Environment Sustainable Development, MIS 5033216.

**Institutional Review Board Statement:** Not applicable.

**Data Availability Statement:** The data supporting the reported results of this study can be provided upon request to the last author.

**Acknowledgments:** The authors want to thank the staff of the Management Unit of Northern Pindos National Park (N.E.C.C.A.), especially Antonios Stagogiannis and Athanassia Karambina, for their involvement in the monitoring schemes, and the Public Power Corporation S.A. for providing their boat and assisting in the fish sampling.

**Conflicts of Interest:** The authors declare no conflict of interest.

## Appendix A

**Table A1.** Estimated parameters of the length–weight relationship for pumpkinseed *Lepomis gibbosus* in European freshwater systems.

| Region | Country | Length Type | Length Range | a | b | $R^2$ | Year | Reference |
|--------|---------|-------------|--------------|---|---|-------|------|-----------|
| Aoos Springs | Greece | TL | 24–101 | 0.012 | 3.190 | 0.985 | 2021 | Present study |
| River Vit | Bulgaria | SL | 33–92 | 0.024 | 3.187 | 0.963 | 2008 | [65] |
| River Segura | Spain | TL | 48–88 | 0.015 | 2.980 | 0.994 | 2000–2004 | [37] |
| Lake Bara | Croatia | TL | | 0.013 | 3.140 | 0.991 | 2015 | [66] |
| Catchment Basin | Turkey | TL | 27–107 | 0.008 | 3.491 | 0.950 | 2014 | [38] |
| Catchment Basin | Turkey | TL | 39–132 | 0.013 | 3.138 | 0.987 | 2014 | [38] |

**Table A2.** Growth parameters of the species pumpkinseed *L. gibbosus* in freshwater systems in Europe, North America, and Canada. Formatted table of [22].

| Region | Country | L∞ | k | $t_0$ | $t_{max}$ | $\varphi'$ | $r^2$ | Year |
|--------|---------|-----|---|-------|-----------|-----------|-------|------|
| | | Europe | | | | | | |
| Artificial Lake of Aoos (Present study) | Greece | 119 | 0.36 | −0.31 | 8 | 3.707 | 0.691 | 2021 |
| Tapada Pequena Reservoir | Portugal | 81.3 | 0.41 | −0.151 | 7 | 3.43 | 0.995 | 1996 |
| Tapada Grande Reservoir | Portugal | 112.5 | 0.16 | −1.482 | 17 | 3.32 | 0.897 | 2004 |
| Monte Novo Reservoir | Portugal | 101.1 | 0.50 | −0.108 | 6 | 3.70 | 0.998 | 2004 |
| Divor Reservoir | Portugal | 91.1 | 0.43 | −0.550 | 6 | 3.56 | 0.989 | 1989 |
| Artificial Lake Kerkini | Greece | 163.5 | 0.13 | −2.729 | 20 | 3.53 | 1.000 | 1994 |
| Lake Banyoles | Spain | 131.8 | 0.57 | 0.367 | 6 | 3.99 | 0.998 | – |
| Fumemorte Canal | France | 113.6 | 0.34 | 0.013 | 9 | 3.65 | 0.997 | 2001 |
| Sollac Marsh | France | 133.5 | 0.36 | 0.029 | 8 | 3.80 | 0.997 | 2001 |
| Dabas Pond | Hungary | 94.9 | 0.38 | −0.059 | 8 | 3.54 | 0.996 | 1977 |
| Danube River | Slovakia | 168.3 | 0.23 | −0.158 | 13 | 3.82 | 1.000 | 1973 |

**Table A2.** *Cont.*

| Region | Country | L∞ | k | $t_0$ | $t_{max}$ | $\varphi'$ | $r^2$ | Year |
|---|---|---|---|---|---|---|---|---|
| Cottesmore Pond | England | 94.5 | 0.19 | −0.721 | 15 | 3.23 | 0.996 | 2002 |
| Odra River | Poland | 162.7 | 0.31 | −0.584 | 9 | 3.92 | 0.994 | − |
| North America | | | | | | | | |
| Deep Lake | Michigan, USA | 190.7 | 0.17 | 0.048 | 18 | 3.78 | 0.990 | 1948 |
| Lower Dowsley Pond | Ontario, CAN | 174.2 | 0.24 | 0.195 | 13 | 3.87 | 1.000 | 1990 |
| Opinicon Lake | Ontario, CAN | 184.5 | 0.14 | −0.647 | 21 | 3.67 | 0.994 | 1990 |
| Upper Poole Pond | Ontario, CAN | 145.5 | 0.21 | −0.212 | 14 | 3.65 | 0.996 | 1990 |
| Houghton Lake | Michigan, USA | 256.1 | 0.20 | 0.473 | 15 | 4.11 | 0.990 | 1926 |
| Flora Lake | Wisconsin, USA | 205.5 | 0.16 | −0.327 | 18 | 3.84 | 0.993 | 1958 |
| Plastic Lake | Ontario, CAN | 180.0 | 0.16 | 0.124 | 19 | 3.74 | 0.970 | 1986 |
| Crosson Lake | Ontario, CAN | 168.0 | 0.17 | 0.152 | 18 | 3.68 | 0.986 | 1986 |
| Lake Upper Beverly | Ontario, CAN | 219.0 | 0.13 | −0.026 | 23 | 3.78 | 0.992 | 1987 |
| Lake Lower Beverly | Ontario, CAN | 125.3 | 0.26 | −0.147 | 11 | 3.62 | 0.991 | 1987 |
| Lake Vert | Quebec, CAN | 153.0 | 0.16 | −0.054 | 19 | 3.58 | 0.982 | 1979 |
| Lake Lower Loch Alpine | Michigan, USA | 142.8 | 0.18 | −0.612 | 16 | 3.56 | 0.989 | 1977 |
| A number of 55 lakes | Wisconsin, USA | 195.7 | 0.16 | −0.588 | 18 | 3.79 | 0.995 | 1992 |
| A number of 14 water bodies | Pennsylvania, USA | 160.1 | 0.23 | 0.393 | 13 | 3.78 | 0.982 | 1977 |
| Unnamed Pond | Michigan, USA | 130.4 | 0.24 | −0.802 | 12 | 3.62 | 0.997 | 1938 |
| Unspecified Lake | North Carolina, USA | 280.5 | 0.15 | 0.472 | 20 | 4.07 | 0.981 | 1997 |
| Unspecified Lake | New York, USA | 175.5 | 0.24 | 0.386 | 13 | 3.86 | 0.995 | 1977 |
| A number of 26 ponds | Rhode Island, USA | 172.6 | 0.27 | 0.247 | 11 | 3.91 | 0.999 | 1977 |
| Multiple water bodies | Delaware, USA | 154.0 | 0.63 | 0.287 | 5 | 4.17 | 0.997 | 1997 |
| Multiple streams | North Carolina, USA | 237.0 | 0.17 | 0.312 | 18 | 3.98 | 0.978 | 1977 |
| Multiple water bodies | Illinois, USA | 175.8 | 0.20 | −1.639 | 13 | 3.79 | 0.988 | 1977 |
| Multiple water bodies | Ohio, USA | 189.0 | 0.24 | 0.289 | 13 | 3.93 | 0.999 | 1977 |
| Multiple water bodies | Minnesota, USA | 213.4 | 0.19 | 0.014 | 16 | 3.94 | 0.999 | 1977 |
| Multiple water bodies | Minnesota, USA | 292.5 | 0.17 | 0.211 | 18 | 4.16 | 0.981 | 1977 |

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
