# Peer review of "Notes on the Summer Life History Traits of the Non-Native Pumpkinseed (Lepomis gibbosus) (Linnaeus, 1758) in a High-Altitude Artificial Lake"

_diversity, doi:10.3390/d15080910_

Round 1

Reviewer 1 Report

The manuscript brings important data on the biology of invasive fish in suboptimal habitat. The analysis is well done and the results could be interesting to the broad scientific community but there are several issues with the manuscript. 

The aims are not conclusive in the introduction and in the discussion, the introduction needs thorough editing and rewriting, and GSI analysis is questionable due to the short sampling period.

Specific comments are listed in the Word document.

I would suggest employing a native speaker to edit the language because in some places is hard to understand what you are stating, which impedes the manuscript's quality. 

Author Response

We would like to thank Reviewer 1 for his/her fruitful comments and suggestions that highly improve the quality and chariness of our manuscript. In general, the text has been revised by an English native speaker and special attention was given to the long sentences, which have been broken down into more comprehensive units. Also, the sections of the introduction and discussion have been re-structured according to the reviewers’ comments. Final, the term of “impoundment of Aoos springs” has been replaced with the term “artificial lake of Aoos” throughout the text. At the attached file, we reply to his/her comments accordingly. Also, changes are pointed with track changes within the text.

Reviewer 2 Report

Invasive fish species have long been mastering the reservoirs of Europe. The biology of each invasive species may differ depending on the conditions of the reservoir. However, such studies are necessary to understand the directions of the spread of fish invasions in the future.

- You have used electro-fishing at 4 stations. Give a description of each station (description of the bottom, macrophytes, depth, and more).

- There is no section on statistical processing of the material in the methods. Write it and specify all statistical methods.

- There is no conclusion. The authors should write a conclusion.

Author Response

We would like to thank Reviewer 2 for his/her fruitful comments and suggestions that highly improve the quality and chariness of our manuscript. In general, the text has been revised by an English native speaker and special attention was given to the long sentences, which have been broken down into more comprehensive units. Also, the sections of the introduction and discussion have been re-structured according to the both reviewers’ comments.
